# Intramedullary Nailing with and without the Use of Bone Cement for Impending and Pathologic Fractures of the Humerus in Multiple Myeloma and Metastatic Disease

**DOI:** 10.3390/cancers15143601

**Published:** 2023-07-13

**Authors:** Andriy Kobryn, Patrick Nian, Joydeep Baidya, Tai L. Li, Aditya V. Maheshwari

**Affiliations:** Department of Orthopaedic Surgery and Rehabilitation Medicine, State University of New York Downstate Health Sciences University, Brooklyn, NY 11203, USA

**Keywords:** intramedullary nailing, bone cement, humerus, multiple myeloma, metastatic disease

## Abstract

**Simple Summary:**

Although intramedullary nailing (IMN) is considered the standard of care for the surgical management of most femur metastatic diseases, the optimal treatment of metastatic humeral impending and/or pathologic fractures is still debatable. In this study, we explored the usage of cemented vs. uncemented IMN in treating both impending and pathologic fractures, secondary to metastatic disease or multiple myeloma, and compared the outcomes in terms of survival, function, blood loss, blood transfusions, and perioperative complications. Our findings demonstrated that both groups had comparable outcomes, except higher blood loss was found in the cemented group. Thus, intramedullary nailing, both with and without cement, is a relatively safe and effective therapeutic modality for metastatic humeral disease in select patients with similar clinical outcomes and acceptable complication rates.

**Abstract:**

Although intramedullary nailing (IMN) is considered the standard of care for the surgical management of most femur metastatic diseases, the optimal treatment of metastatic humeral impending and/or pathologic fractures is still debatable. Moreover, the use of cemented humeral nails has not been thoroughly studied, and only a few small series have compared their results with uncemented nails. The purpose of this study was to compare the (1) survivorship, (2) functional outcomes, and (3) perioperative complications in patients receiving cemented versus uncemented humerus IMN for impending or complete pathologic fractures resulting from metastatic disease or multiple myeloma. We retrospectively reviewed 100 IMNs in 82 patients, of which 53 were cemented and 47 were uncemented. With a mean survival of 10 months (Cemented: 8.3 months vs. Uncemented: 11.6 months, *p* = 0.34), the mean Musculoskeletal Tumor Society (MSTS) scores increased from 42.4% preoperatively (Cemented: 40.2% vs. Uncemented: 66.7%, *p* = 0.01) to 89.2% at 3 months postoperatively (Cemented: 89.8% vs. Uncemented: 90.9%, *p* = 0.72) for the overall group (*p* < 0.001). Both cohorts yielded comparable complication rates (overall [22.6% vs. 19.1%)], surgical ([11.3% vs. 4.3%], and medical [13.2% vs. 14.9%], all *p* > 0.05), but estimated blood loss was significantly higher in the cemented group (203 mL vs. 126 mL, *p* = 0.003). Thus, intramedullary nailing, with and without cement augmentation in select patients, is a relatively safe and effective therapeutic modality for metastatic humeral disease with similar clinical outcomes and acceptable complication rates. While controlling for possible selection bias, larger-scale, higher-level studies are warranted to validate our results.

## 1. Introduction

The skeletal system represents the third most common site for metastases, following the lungs and liver, with the humerus being the second most affected long bone (after the femur), accounting for about 20% of all metastatic long bone lesions [1]. Bone metastases can present as impending or complete pathologic fractures that usually occur late in the course of the disease, with median survivorship falling below 1 year once diagnosed [2,3,4]. Given the unpredictable healing process of pathological fractures, the involvement of multiple bones, adjuvant therapy requirements, and early rehabilitation issues, non-operative treatments have yielded unsatisfactory results, and surgical stabilization has become the recommended therapeutic strategy [2,4,5,6,7,8].

Although intramedullary nailing (IMN) is widely considered the surgical standard of care for most occurrences of metastatic bone disease in the femur, there is still vast debate on the optimal treatment for the humerus [3,4,5,7,8]. Surgical options for the humerus include IMN, plate/screws, diaphyseal prostheses, and arthroplasty. The treatment is influenced by multiple factors, including lesion location and severity, bone quality, presence of complete fracture, and overall patient health [2,9,10,11]. Diaphyseal fractures are most frequently treated with IMN, plate–screw constructs, or diaphysis prostheses, whereas endo-prosthetic reconstruction or IMN with a supplemental plate are often used for periarticular humeral lesions [12,13,14,15]. In addition to protecting the entire humeral length, IMN is associated with less soft-tissue dissection and blood loss, shorter operative time, lower complication rates, and earlier rehabilitation [2,4,5,7,9,13,16,17,18]. However, concerns regarding IMN placement remain as complications, such as disease progression, neurovascular and rotator cuff damage, fat and tumor emboli, and fixation loss, are prevalent [10,19]. Additionally, limited open or closed IMN techniques often leave behind the tumor burden and preclude the use of the local adjuvant options of radio-resistant tumors, such as cryosurgery, argon beam, and phenol therapy. On the other hand, plating, prostheses, and arthroplasty may not protect the whole bone and may lead to more soft-tissue dissection, higher blood loss, increased frequency of complications, prolonged functional recovery, and delayed adjuvant chemotherapy or radiation [11,17].

The use of bone cement with the nail has been shown to promote fixation, provide greater construct stability, reduce local tumor mass, resume early adjuvant treatment, and slow disease progression [2,14,20,21]. However, some consider cemented IMN for humerus to be unnecessary due to increased complications, operative time, adequate stability with the nail alone, possible neurovascular damage secondary to the cement’s thermogenic effect and soft-tissue extrusion, and slower bony healing [2,16,22,23]. As of yet, the superiority of any of these modalities has not been established, and there is a paucity of studies that directly compare the survivorship, functional outcomes, and perioperative complications between patients undergoing cemented and uncemented humeral IMN (Appendix A) [2,4,5,8,11,14,15,16,17,18,20,21,24,25,26,27,28,29,30,31,32,33,34,35,36,37,38,39,40,41]. Moreover, many of these studies have not distinguished if the bone cement was only used to fill the tumor cavity or the entire intramedullary canal (cemented IMN, as defined in our study).

Therefore, the current analysis aimed to study the effectiveness of IMN as a treatment for impending and pathologic humeral fractures in metastatic bone disease and multiple myeloma by assessing (1) survivorship, (2) functional outcomes, and (3) perioperative complication rates for the whole group, as well as by comparing the cemented IMN vs. uncemented IMN groups.

## 2. Materials and Methods

### 2.1. Patient Selection and Demographics

The present study was a retrospective comparative analysis of a prospectively maintained institutional review board-approved single-surgeon database in an urban academic setting. Between August 2011 and July 2022, a total of 82 patients (41 [50%] males and 41 [50%] females) who underwent IMN for impending or complete pathologic humeral fractures, for metastatic disease (*n* = 28) or multiple myeloma (*n* = 54), were included. Some of these patients have been described, earlier, in a separate study that investigated the outcomes of single stage multiple nailing procedures [42]. Among these patients, 18 underwent bilateral IMN (8 cemented vs. 10 uncemented), bringing the total number of humeral nails placed to 100 (53 [53%] cemented and 47 [47%] uncemented) (Figure 1). There were 40 (40%) and 60 (60%) nails placed for impending and complete pathologic fractures, respectively (Table 1). Out of 82 patients, 33 (40.2%) (cemented: 26 [56.5%] vs. uncemented: 7 [19.4%]) had isolated the humeral nails placed, while 49 (59.8%) (cemented: 20 [43.5%] vs. uncemented: 29 [80.6%]) had more than 1 nail placed in other long bones in 1 or more settings. The most common combination was unilateral humerus and unilateral femur IMN (18; 8 cement and 10 uncemented), followed by unilateral humerus and bilateral femora IMN (9; 2 cement and 7 uncemented).

The demographics, except for the following, were similar in both groups (Table 1). The uncemented group had more impending fractures than the cemented group, who had more completed pathologic fractures. The uncemented group had more diaphyseal lesions, and the cemented group had more proximal lesions. The uncemented group had more patients receiving multiple nails placed in one setting. Within the patients who received multiple nails, 38 (cemented: 15 [32.6%] vs. uncemented: 23 [63.9%]) were performed in one setting, while 11 (cemented: 5 [10.9%]) vs. uncemented: 6 [16.7%]) were performed in 2 or more settings.

### 2.2. Perioperative Protocols and after Care

All patients received coordinated care from a multidisciplinary team, ensuring medical optimization prior to surgery, including a hemoglobin level of at least 10 g/dL. Preoperative embolization of hyper-vascular metastatic tumors was performed in 2 cases (renal cell and hepatocellular carcinomas). Mirels’ criteria were mostly used for the prophylactic fixation of impending fractures [43]. The type of IMN (cemented vs. uncemented) predominantly depended on the extent of tumor involvement, bone quality, amount of bone loss, periarticular lesions, extensive and skip lesions, and fixation stability, and it was decided by the senior surgeon (AVM).

The surgical technique is detailed in Appendix A [44]. All surgeries were conducted with a minimally invasive technique, except for 7 pathologic fractures that required direct tumor site opening (curettage resection for radioresistant tumors and/or significant bone defects [5 cases], as well as failure to obtain acceptable fracture closed reduction [2 cases]). There were 3 patients who received plate fixation, in addition to intramedullary nailing, for extensive periarticular bone involvement. In addition, one patient in the cemented cohort required intraoperative conversion from a nail to a plate construct due to a proximal nail cutout of the lateral cortex. This patient was excluded from functional score analysis, but they were included in survival and complication analyses. The postoperative rehabilitation protocol was similar for all patients. An arm sling was given for initial postoperative comfort, and immediate range of motion, starting with pendulum exercises, was allowed. Weight bearing depended on bone involvement and fixation stability, but activities of daily living were allowed immediately. After discharge, outpatient visits were scheduled at 2 and 6 weeks postoperatively, every 3 months for the first year, every 6 months for the second year, and once a year afterwards. Radiographs were obtained immediate postoperatively, at 3 months, and at every subsequent visit (Figure 2 and Figure 3). In most patients, chemotherapy and/or radiation (3000–3500 Gy) were started/resumed 10 to 21 days after their index surgery, as deemed appropriate by the multidisciplinary team.

### 2.3. Collected Variables

Extracted variables included patient demographics (age, sex, body mass index [BMI]), primary malignancy diagnosis, fracture type (impending or complete pathologic), lesion location, cement use, concomitant procedures (e.g., other long bone IMN in the same or different setting), intraoperative blood loss, and blood transfusion volume (up to 24 h postoperatively, 1-unit PRBC = 325 mL). Perioperative complications and the return to the operating room were recorded. Complications were labeled as “surgical” if they were directly related to the nailing procedure, and they included infection, wound dehiscence, reoperations, and implant-related mechanical complications. Some cement extrusion to soft tissues was common in pathologic fracture cases, but it was only classified as a complication if there were subsequent mechanical issues, neurovascular damage, and/or a need for additional intervention. Medical complications were primarily recorded for that admission and cardiopulmonary, including pulmonary embolism, pneumonia, hypotension, myocardial infarction, and respiratory distress. Other recorded complications included gastrointestinal bleeding, Clostridium difficile infection, coagulopathies such as disseminated intravascular coagulation, deep vein thrombosis, and thrombocytopenia, sepsis, urinary tract infection, as well as transaminitis with hyperbilirubinemia. Intraoperative estimated blood loss (EBL) was estimated by quantifying the number of laparotomy sponges utilized along with the total blood in the suction canister [45]. Functional outcomes were recorded using the Musculoskeletal Tumor Society (MSTS) upper extremity scoring system at the initial presentation and 3 months postoperatively [46]. Due to death, loss to follow-up, or an inability to have a direct follow-up visit, especially during the COVID-19 pandemic, functional score was not available for the majority of patients beyond 3 months, and thus, the MSTS score was reported for 3 months. Oncologic outcomes were evaluated with patient survivorship and calculated by tracking patients from surgery until death. Several of our patients were foreign citizens traveling to and from their home country to receive treatment, making data collection difficult for certain variables. Patient death was documented from medical chart review and direct communication with their families and other medical providers [47]. When mortality was not able to be confirmed by available means and no further communication with the patient or their family members was identified in the patient’s chart, individuals who missed two consecutive visits without rescheduling were presumed to be lost to follow-up at the date of the last documented visit to the clinic or hospital, and they were subsequently excluded from analysis of survivorship and functional outcomes. By a 3 month postoperative period, 18 (22.0%) patients died and 18 (22.0%) were lost to follow-up. By a 6 month postoperative period, 23 (28.0%) patients died and 25 (30.5%) were lost to follow-up. By a 12 month postoperative period, 27 (32.9%) patients had died and 29 (35.4%) were lost to follow-up, and thus, analysis was done based on available patients at each time point.

### 2.4. Statistical Analyses

Descriptive analyses were performed to summarize patient demographics, operative variables, and patient outcomes. Statistical analyses included two-sided Fisher’s exact and Student’s *t*-tests to compare categorical and continuous variables, respectively. Patient survivorship was reported using Kaplan–Meier estimates with the log–rank test. All analyses were performed in R Statistical Software using a *p*-value of <0.05 as the threshold for statistical significance.

## 3. Results

A total of 82 patients with 100 nails were analyzed (53 [53.0%] cemented and 47 [47.0%] uncemented).

### 3.1. Survivorship

Mean survivorship for all available patients was 10.0 ± 14.3 (range, 0–86) months. No significant differences in mean survivorship were detected between the two cohorts (cemented: 8.3 ± 9.3 [range, 0–35] vs. uncemented: 11.6 ± 17.7 [range, 0–86] months, *p* = 0.34) (Table 2). Kaplan–Meier median survivorship was estimated at 6.0 months for the overall patient population. No significant differences in the Kaplan–Meier median survivorship were detected between the two cohorts (cemented: 6.0 vs. uncemented: 8.0 months, *p* = 0.30) (Figure 4). There was no intraoperative death. There were 2 patients in the cemented group who died during the same admission, at 4 weeks, compared to none in the uncemented group (*p* = 0.22; Table 2 and Table 3).

### 3.2. Functional Outcomes

Mean MSTS scores for all available patients increased from 42.4% ± 8.4% (range, 28–60%) preoperatively to 89.2% ± 5.5% (range, 76–96%) at 3 months postoperatively (*p* < 0.001). Patients with cemented IMN had lower preoperative MSTS scores (40.2% ± 9.6% [range, 16–64%]) than those without cement (66.7% ± 28.5% [range, 20–100%], *p* = 0.01). However, postoperative MSTS scores were comparable between both groups (cemented: 89.8% ± 7.0% [range, 80–100%]) vs. uncemented: 90.9% ± 1.9% [range, 90–100%]) (*p* = 0.72) (Table 2).

### 3.3. Perioperative Outcomes and Complications

The details of the complications are summarized in Table 2 and Table 3. There was a total of 21 (21.0%) perioperative complications in 100 surgeries, of which 14 (14.0%) and 8 (8.0%) were medical and surgical, respectively. There was one cemented IMN patient who had both medical and surgical complications. In the cemented IMN cohort, there were 12 (22.6%) perioperative complications in 53 surgeries, of which 7 (13.2%) were medical and 6 (11.3%) were surgical. In the uncemented IMN cohort, there were 9 (19.1%) perioperative complications in 47 surgeries, of which 7 (14.9%) were medical and 2 (4.3%) were surgical. Overall (22.6% vs. 19.1%; *p* = 1.00), medical (13.2% vs. 14.9%; *p* = 0.76), and surgical (11.3% vs. 4.3%; *p* = 0.47) complication rates were similar between patients undergoing cemented and uncemented humeral IMN, respectively. There were no cases of surgical site infections, wound dehiscence, or implant breakage requiring returns to the operating room.

There were two returns to the operating room in the entire study. From the cemented group, 1 patient developed quadriparesis from cervical spine tumor involvement and instability, requiring urgent neurosurgical decompression and fixation 2 days after nailing procedure. Another individual from the cemented group was taken back to the operating room, 2 years after their index surgery, for a symptomatic prominent backed out proximal locking screw that was removed.

In the entire study cohort, there was a mean EBL of 166 ± 138 (range, 0–1000) mL (Table 2). EBL was significantly higher in patients with cement-augmented IMN (203 ± 179 [range, 0–1000] mL) compared to patients with uncemented IMN (126 ± 49 [range, 50–300] mL) (*p* = 0.003). There was a mean perioperative transfusion of 236 ± 394 (range, 0–1950) mL for both study cohorts (Table 2). There were no significant differences in perioperative transfusion volumes between the cemented (204 ± 318 [range, 0–975] mL) and uncemented (271 ± 462 [range, 0–1950] mL).

## 4. Discussion

Patients presenting with impending or complete pathologic humeral fractures, secondary to metastatic bone disease or myeloma, may be treated with multiple modalities. Although IMN fixation is commonly used, the procedure has been associated with various complications [2,4,5,7,9,10,13,16,17,18,19]. Furthermore, the benefit of IMN, with or without cement, has not been adequately addressed in the literature, and most studies have not compared clinical outcomes [2,4,5,8,11,14,15,16,17,18,20,21,24,25,26,27,28,29,30,31,32,33,34,35,36,37,38,39,40,41].

In our study population, median and 1 year survival were comparable to those previously reported by other studies (6.0 vs. 6.4–10.6 months and 38.7% vs. 26.7–49%, respectively), although there may be a difference in calculation methodology and follow-ups between studies, impeding direct comparison [2,14,17,21,27,49,50]. In our study, there was no difference in the survival of patients in the cemented vs. uncemented group. With the numbers available, this may signify that survival after metastasis to the humerus has a similar prognosis, irrespective of the extent of the disease. Thus, surgical management mainly stays a part of palliative care, improving pain, function, and quality of life.

Functional outcomes of patients with metastatic humeral disease improved following IMN insertion, as evidenced by a two-fold overall mean MSTS score improvement at 3 month follow-up. Coupled with the relatively low surgical complication rate and adequate functional gain, our study found IMN fixation to be an effective operative modality for our patients. Patients undergoing cemented humeral IMN exhibited significantly lower preoperative MSTS scores (40.2%) than patients with uncemented nail insertion (66.7%) (*p* = 0.01). While this was likely due to selection bias, both groups were similar with respect to postoperative MSTS scores at 3 months (89.8% and 90.9%), justifying the use of cement in this group. Given that individuals with larger tumor burdens—and resultant poorer bone stock—are more likely to receive cement augmentation (over 70% of pathologic fractures in the current analysis), bone cement may promote functional gain in more advanced humeral diseases. This finding corroborates previous studies that endorse cement for the improvement of fixation stability, pain mitigation, and function restoration, especially in the short term [10,20,51,52,53].

Perioperative complications occurred in 21.0% of procedures, with medical (14.0%) and surgical (8.0%) complication rates comparable to those reported in the literature (0–26.0% and 0–28.6%, respectively) [Table 2 and Table 3, Appendix A]. The variation in complication rates between studies could be explained by discrepancies in defining adverse events. For instance, while some analyses placed emphasis, mostly, on cardiopulmonary complications, including thromboembolism, others also included gastrointestinal bleeding and postoperative pneumonia as complications [17,18]. Embolic events are notable in IMN fixation because reaming, nail insertion, and cement augmentation may cause an intramedullary pressure surge, potentially increasing the risk of tumor and bone marrow fat liberation, subsequently leading to an intraoperative cardiopulmonary event [28,54]. However, this has, mostly, been studied in the femur and not the humerus. In this study, only 1 (1.0%) case of symptomatic pulmonary embolism was noted, which is consistent with a previously reported incidence of fat (0.3%) and pulmonary (1.3%) emboli following humeral nail insertion [17]. In our patients undergoing cemented IMN, cement was used in a relatively more liquid state without pressurization, which, combined with predominant use of unreamed or minimally reamed smaller diameter (8 mm) nails and the presence of a fracture acting as a natural vent site, could have contributed to the overall low rate of embolic events.

Patients undergoing cemented and uncemented humeral IMN displayed comparable rates of overall (22.6% vs. 19.1%), medical (13.2% vs. 14.9%), and surgical (11.3% vs. 4.3%) complications. Intraoperative surgical complications (4 [4.0%] cases) primarily occurred in individuals treated with cemented IMN, reflecting the higher proportion of pathologic fractures in this patient population, more extensive disease with relatively poorer bone stock, and the procedure’s technical complexity. Although our cementing technique increased the probability of extrusion at the fracture sites, many of these at the fracture site had no clinical significance and required no intervention, possibly due to an intact protective soft-tissue sleeve, as most of these fractures are usually low-energy events. When needed, the cement was removed in the same setting by using either the same or a different incision. Adding to the literature, most surgical complications encountered with cemented nails were technical, which were recognized and managed appropriately with no impact on patient outcomes [21,29]. Furthermore, intraoperative blood loss was also higher in subjects with cemented IMN, which usually required taking additional surgical steps, increasing canal reaming, and, possibly, lengthening the operative time as a result, although this extra time was slightly offset by not using distal interlocking screws.

The current study has several limitations. Given its retrospective and complex nature, no true matched control group was included. As such, there were certain confounders of survivorship, functional outcomes, and perioperative complications that could not be accounted and controlled. Several patients underwent multiple long bone nailing procedures, either in a single stage or multiple staged fashion, which can influence overall outcomes. Moreover, there was a selection bias for the usage of cement based on the surgeon’s assessment. Long bones requiring prophylactic fixation have relatively better quality, so uncemented intramedullary nails were, more often, inserted for diaphyseal lesions and impending fractures. Diaphyseal lesions also have better proximal and distal bones remaining for fixation and interlocking screw purchasing, thus precluding the need for cement augmentation. Patients with uncemented nails also underwent significantly more than one IMN procedure in multiple long bones in the same setting, creating another bias for complications, as anticipated complexity, surgical time, and blood loss was less. In contrast, complete pathologic fractures often stem from more advanced disease, and as a result, they exhibit inferior bone quality, warranting cement augmentation for better fixation. The lower preoperative MSTS score in subjects who underwent cemented nailing could have mirrored the higher percentage of complete pathologic fractures documented in this patient population. Despite evaluating one of the largest groups of a detailed humeral IMN reported to date (Appendix A), the sample size included in the current analysis is still relatively small, and it may be limited by insufficient statistical power for the comparison of the two cohorts. Moreover, several patients were lost to follow-up, and they were excluded from the analysis. This study also did not compare operative time directly, as overall surgical time was influenced by several independent factors such as fluoroscopy, perioperative additional anesthesia preparation, and multiple single-stage nailing [42]. Likewise, we did not compare the length of hospital stays and returns to definitive adjuvant therapy in this patient population due to several factors listed above. In addition, most subjects had multiple myeloma as their primary malignancy, due to the disease’s high prevalence in our community. Nevertheless, there was no significant difference (*p* > 0.05) in perioperative complications or survivorship between those with multiple myeloma and those with metastatic disease. Despite similar orthopedic management, metastatic tumors and multiple myeloma may differ regarding disease history and prognosis, potentially skewing the study’s findings.

To the best of our knowledge, this study is the largest series of its kind, despite these limitations, and it is the first to compare the survivorship, functional outcomes, and perioperative complications of cemented and uncemented IMN in order to assess their effectiveness as treatments for impending and complete pathologic humeral fractures. Future plans may include a higher-powered analysis, an in-depth analysis of differences between patients with multiple myeloma and metastatic disease, as well as a way to account for other variables and confounders.

## 5. Conclusions

Intramedullary nailing, both with and without cement, is a relatively safe and effective therapeutic modality for impending and pathologic humeral fractures, resulting in similarly acceptable clinical outcomes and complication rates. The use of bone cement is often based on several clinical factors and, thus, induces selection bias. Nevertheless, the outcomes were similar between both cemented and uncemented groups, justifying the use of cement in this select group. Most intraoperative surgical complications resulted from technical errors stemming from bone cement use, and they could be minimized with awareness, meticulous attention to surgical technique, and more abundant experience. While controlling for possible selection bias, larger-scale higher-level studies are warranted to validate these results.

## Figures and Tables

**Figure 1 cancers-15-03601-f001:**
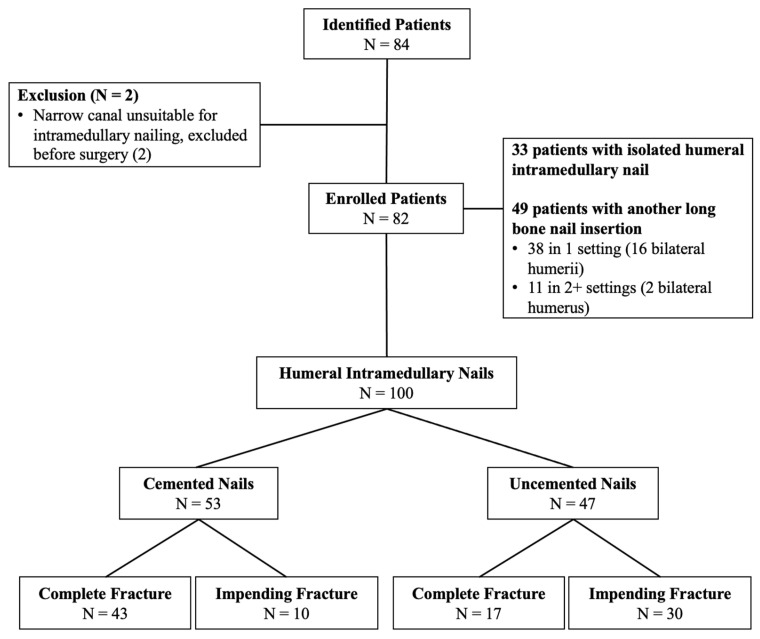
Flow chart depicting the patient selection process adopted in the current investigation. There were 82 subjects enrolled, accounting for 100 intramedullary nailing procedures.

**Figure 2 cancers-15-03601-f002:**
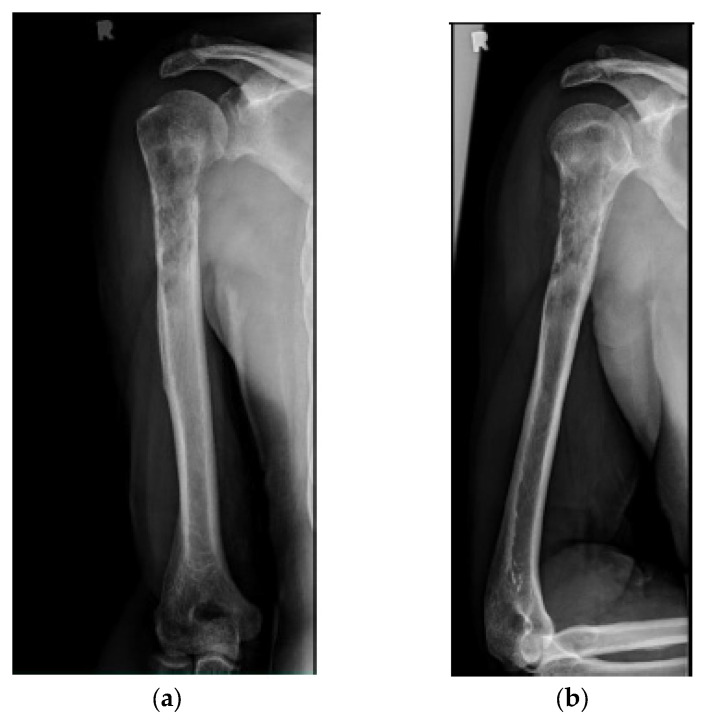
(**a**) AP and (**b**) lateral right humerus radiographs of a 67-year-old male with a mixed lytic sclerotic lesion in the proximal meta-diaphyseal region, with a pathologic fracture from a newly diagnosed metastatic prostate cancer. This was treated by a bone biopsy, followed by a cemented IMN with two proximal inter-locking screws and no distal screw, as shown in the (**c**) AP and (**d**) lateral humerus radiographs. Cement was used for augmentation, due to poor proximal humerus bone quality, to support the nail and the inter-locking screws.

**Figure 3 cancers-15-03601-f003:**
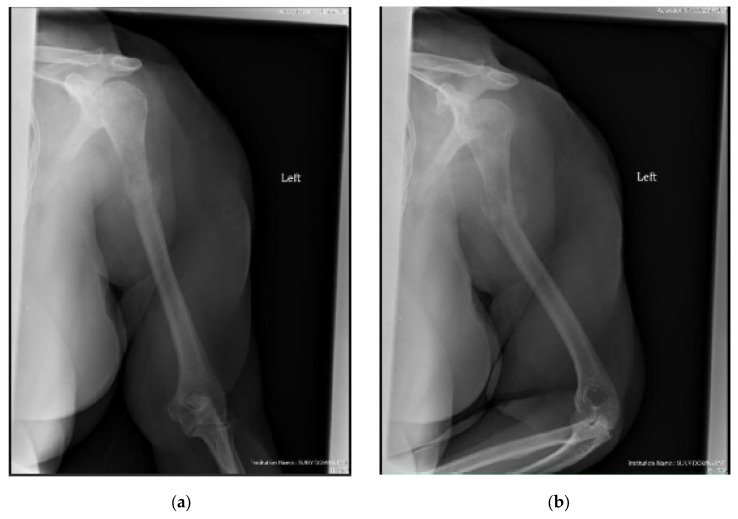
(**a**) AP and (**b**) oblique left humerus radiographs of a 47-year-old female with mixed lytic sclerotic lesion in the proximal meta-diaphyseal region, with a pathologic fracture and periosteal reaction in a patient with established metastatic breast cancer. This was treated by an uncemented IMN, with three proximal locking screws and one distal screw, as shown in (**c**) AP and (**d**) oblique humerus radiographs. No cement was used, as there was enough proximal and distal bone to support the nail and the inter-locking screws, and some healing changes were already evident.

**Figure 4 cancers-15-03601-f004:**
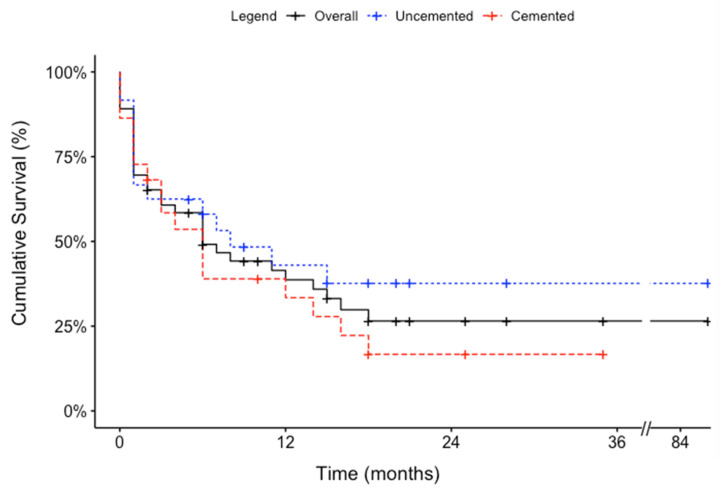
Kaplan–Meier curves depicting the survival of patients who underwent humeral intramedullary nail fixation with and without cement.

**Table 1 cancers-15-03601-t001:** Demographic variables in the entire study cohort and in patients by cement usage.

	Entire Patient Population	Cemented Cohort	Uncemented Cohort	*p*-Value ^ǁ^
	N = 82 (100%)	N = 46 (100%)	N = 36 (100%)	
Age (Year) *	68.0 ± 11.7 (38–96)	67.9 ± 10.9 (38–90)	68.3 ± 13.3 (44–96)	0.46
Body Mass Index (Kg/m^2^) *	27.6 ± 5.6 (17.1–45.2)	27.2 ± 5.1 (17.2–37.6)	28.2 ± 6.2 (21.4–45.2)	0.07
Sex ^§^				1.00
Male	41 (50.0%)	23 (50.0%)	18 (50.0%)	
Female	41 (50.0%)	23 (50.0%)	18 (50.0%)	
Concomitant Procedure	49 (59.8%)	20 (43.5%)	29 (80.6%)	1.00
Bilateral Humerii	18 (22.0%)	8 (17.4%)	10 (27.8%)	
Another Long Bone	31 (37.8%)	12 (26.1%)	19 (52.8%)	
Staging of Procedures				<0.001
Single Nail	33 (40.2%)	26 (56.5%)	7 (19.4%)	
Multiple Nails-One Setting	38 (46.3%)	15 (32.6%)	23 (63.9%)	
Multiple Nails-Two+ Settings	11 (13.4%)	5 (10.9%)	6 (16.7%)	
	**All Nails**	**Cemented Nails**	**Uncemented Nails**	***p*-Value** **^ǁ^**
	**N = 100 (100%)**	**N = 53 (100%)**	**N = 47 (100%)**	
Laterality ^§^				0.49
Left	48 (48.0%)	26 (49.1%)	22 (46.8%)	
Right	52 (52.0%)	27 (50.9%)	25 (53.2%)	
Fracture Type ^§,+^				<0.01
Impending	40 (40.0%)	10 (18.9%)	30 (63.8%)	
Complete	60 (60.0%)	43 (81.1%)	17 (36.2%)	
Major lesion/Fracture Location ^§,†^				0.02
Diaphysis	55 (55.0%)	20 (37.7%)	35 (74.5%)	
Proximal	32 (32.0%)	26 (49.5%)	6 (12.8%)	
Distal	3 (3.0%)	2 (3.8%)	1 (2.1%)	
Diaphysis + Proximal	7 (7.0%)	3 (5.1%)	4 (8.5%)	
Proximal + Distal	1 (1.0%)	1 (1.9%)	0 (0.0%)	
Diaphysis + Proximal + Distal	2 (2.0%)	1 (1.9%)	1 (2.1%)	
Primary Diagnosis ^§^				0.24
Multiple Myeloma	54 (54.0%)	31 (58.5%)	23 (48.9%)	
Breast Cancer	18 (18.0%)	6 (11.3%)	12 (25.5%)	
Prostate Cancer	8 (8.0%)	3 (5.7%)	5 (10.6%)	
Renal Cancer	7 (7.0%)	5 (9.4%)	2 (4.3%)	
Lung Cancer	4 (4.0%)	3 (5.7%)	1 (2.1%)	
Hepatocellular Carcinoma	1 (1.0%)	1 (1.9%)	0 (0.0%)	
Lymphoma	4 (4.0%)	1 (1.9%)	3 (6.4%)	
Melanoma	1 (1.0%)	1 (1.9%)	0 (0.0%)	
Cholangiocarcinoma	1 (1.0%)	1 (1.9%)	0 (0.0%)	
Giant Cell Carcinoma	1 (1.0%)	1 (1.9%)	0 (0.0%)	
Gastrointestinal Stromal Tumor	1 (1.0%)	0 (0.0%)	1 (2.1%)	
Anesthesia Type ^§^				0.20
General Anesthesia	94 (94.0%)	47 (88.7%)	47 (100%)	
Scalene Block	1 (1.0%)	4 (7.5%)	0 (0.0%)	
General Anesthesia + Scalene Block	5 (5.0%)	2 (3.8%)	0 (0.0%)	

* Reported as mean ± standard deviation (range). ^§^ Reported as sample size (%). ^+^ As determined by Mirel’s criteria [43]. ^†^ Although many patients had multiple diffuse lesions, these categories were made based on major lesions that were considered for surgical decision making. ^ǁ^ Reflecting the comparison of variables between patients according to bone cement usage.

**Table 2 cancers-15-03601-t002:** Complications, overall survival, and functional outcomes in study patients.

	All Nails	Cemented Nails	Uncemented Nails	*p*-Value ^ǁ^
	N = 100 (100%)	N = 53 (100%)	N = 47 (100%)	
Medical Complications *	14 (14.0%)	7 (13.2%)	7 (14.9%)	0.76
Surgical Complications *	8 (8.0%)	6 (11.3%)	2 (4.3%)	0.47
Overall Complications *	21 ^+^ (21.0%)	12 ^+^ (22.6%)	9 (19.1%)	1.00
Estimated Blood Loss (mL) ^§^	166 ± 138 (0–1000)	203 ± 179 (0–1000)	126 ± 49 (50–300)	0.003
Single Nail	184 ± 215 (0–1000)	209 ± 238 (0–1000)	100 ± 50 (50–200)	0.046
Multiple Nails-One Setting	149 ± 84 (50–500)	193 ± 110 (100–500)	124 ± 46 (50–300)	0.007
Multiple Nails-Two+ Settings	173 ± 44 (100–200)	183 ± 41(100–200)	164 ± 48 (100–200)	0.45
Perioperative Transfusion (mL) ^§^	236 ± 394 (0–1950)	204 ± 318 (0–975)	271 ± 462 (0–1950)	0.41
Single Nail	115 ± 245 (0–650)	135 ± 270 (0–650)	46 ± 123 (0–325)	0.23
Multiple Nails-One Setting	325 ± 461 (0–1950)	293 ± 348 (0–975)	344 ± 520 (0–1950)	0.67
Multiple Nails-Two+ Settings	175 ± 314 (0–975)	217 ± 394 (0–975)	139 ± 256 (0–650)	0.69
Intraoperative Mortality	0 (0.0%)	0 (0.0%)	0 (0.0%)	1.00
Mortality in the Same Admission	2 (2.0%)	2 (3.8%)	0 (0.0%)	0.22
Preoperative MSTS Score (%) ^†^	42.4 ± 8.4 (28–60)	40.2 ± 9.6 (16–64)	66.7 ± 28.5 (20–100)	0.01
Postoperative MSTS Score at 3 months (%) ^†^	89.2 ± 5.5 (76–96)	89.8 ± 7.0 (80–100)	90.9 ± 1.9 (90–100)	0.72
Survival Time (months) ^†^	10.0 ± 14.3 (0–86)	8.3 ± 9.3 (0–35)	11.6 ± 17.7 (0–86)	0.34

MSTS: Musculoskeletal Tumor Society. * Reported as sample size (%). ^§^ Reported as mean ± standard deviation (range). ^†^ Reported as mean ± standard deviation (range) after excluding patients lost to follow-up. ^+^ One cemented intramedullary nail procedure exhibited a combination of medical and surgical complications. ^ǁ^ Reflecting the comparison of variables between patients according to bone cement usage.

**Table 3 cancers-15-03601-t003:** List of patients undergoing humeral intramedullary nailing who exhibited surgical and medical complications.

Complication Type	Sex *	Age	Primary Diagnosis	Lesion Location	Fracture Type	Setting	Complication	Surgical Intervention	Same Admission Perioperative Mortality
**Cemented**									
Medical	M	77	Multiple myeloma	Proximal	Pathologic ^+^	Postoperative	Gastrointestinal bleeding	No	No
Medical	M	65	Multiple myeloma	Proximal	Pathologic ^+^	Postoperative	Disseminated intravascular coagulation	No	Yes, unresponsive ventricular fibrillation at 4 weeks
Medical	F	38	Breast cancer	Proximal	Pathologic ^+^	Postoperative	Pulmonary embolism	No	No
Medical	M	68	Hepatocellular carcinoma	Proximal	Pathologic ^+^	Postoperative	Sepsis	No	Yes, sepsis and variceal bleeding with disseminated intravascular coagulation at 4 weeks
Medical/ Miscellaneous ^†^	F	60	Multiple myeloma	Diaphysis	Impending	Postoperative	Quadriparesis from cervical metastasis/hypotension/respiratory distress postop day #2	Urgent neurosurgical decompression and fixation. Regained 4 to 4+/5 power ^#^ in 6 months	No
Medical	M	71	Multiple myeloma	Proximal + Distal	Impending	Postoperative	Pneumonia	No	No
Medical + Surgical	F	57	Breast cancer	Proximal + Diaphysis + Distal	Pathologic ^+^	Intraoperative + Postoperative	Cement extrusion into elbow joint ^§^ + Hypotension	Intraoperative elbow arthrotomy and removal using another incision during the same setting	No
Surgical	M	62	Renal cancer	Proximal + Diaphysis	Pathologic ^+^	Intraoperative	Cement extrusion into the shoulder joint as well as broken cement piece in the deltoid soft tissue ^§^	Intraoperative arthrotomy and removal of all extruded cement using another incision during the same setting	No
Surgical	M	87	Prostate cancer	Diaphysis	Pathologic ^+^	Intraoperative	Broken and retained piece of drill bit	No (asymptomatic)	No
Surgical	F	59	Multiple myeloma	Proximal + Diaphysis	Pathologic ^+^	Intraoperative	Proximal nail cutout of the lateral cortex	Intraoperative conversion to plate after aborting nail **	No
Surgical	M	64	Renal cancer	Diaphysis	Pathologic ^+^	Postoperative	High radial nerve palsy	No. Did not have full recovery until death at 4 months with brain metastasis	No
Surgical	M	57	Multiple myeloma	Proximal	Pathologic ^+^	Postoperative	Painful backed out proximal locking screws back out at 2 years	Proximal screws removal two years post index procedure	No
**Uncemented**									
Medical	M	83	Multiple myeloma	Diaphysis	Pathologic ^+^	Postoperative	Hypotension	No	No
Medical	F	55	Breast cancer	Proximal	Pathologic ^+^	Postoperative	Septic shock with thrombocytopenia	No	No
Medical	F	58	Multiple myeloma	Diaphysis	Impending	Postoperative	Transaminitis with hyperbilirubinemia	No	No
Medical	F	68	Multiple myeloma	Diaphysis	Impending	Postoperative	*Clostridium difficile* infection	No	No
Medical	F	71	Multiple myeloma	Proximal Diaphysis	Impending	Postoperative	Pneumonia	No	No
Medical	M	81	Prostate	Diaphysis	Impending	Postoperative	Disseminated intravascular coagulation	No	No
Medical	F	57	Breast cancer	Proximal + Distal	Impending	Postoperative	Hypotension + Respiratory Distress	No	No
Surgical	F	82	Multiple myeloma	Diaphysis	Pathologic ^+^	Postoperative	Proximal nail cutout of the lateral cortex	No (asymptomatic)	No
Surgical	F	44	Breast cancer	Diaphysis	Impending	Postoperative	Partial low radial nerve palsy	No. Full recovery in 3 months	No

* M: Male; F: Female. ^+^ In our study, “pathologic” fracture type is equivalent to a complete pathologic fracture. ^§^ Cement extrusion at the fracture site was common in pathologic fractures, and it was only considered a complication if it required removal either through the same incision or an additional incision. Removal was mainly required if cement extruded into the joint or a cement piece broke, acting as a loose body. ^†^ This patient had a single-stage humerus and femur nailing. Although this complication needed urgent surgical intervention, it was classified as a medical/miscellaneous complication, as it was not directly related to the nailing procedure. Hypotension was only diagnosed/considered when an additional intervention, such as bolus fluids, blood, or medication (pressor), were required. ** This patient was excluded from functional outcome statistical analysis but included for survival and complications analysis. ^#^ Medical Research Council Manual Muscle Testing scale [48].

## Data Availability

The data presented in this study are available on request from the corresponding author. The data are not publicly available due to privacy and ethical considerations.

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
