# Peer review of "Intramedullary Nailing with and without the Use of Bone Cement for Impending and Pathologic Fractures of the Humerus in Multiple Myeloma and Metastatic Disease"

_cancers, 2023, doi:10.3390/cancers15143601_

Round 1

Reviewer 1 Report

This clinical study manuscript demonstrates the effectiveness of intramedullary nailing (IMN) treatment (cemented and uncemented) for impending and pathologic humeral fractures in metastatic bone diseases in different aspects, such as survivorship, functional outcomes, and perioperative complication rates. As IMN is currently considered the gold standard for most large bone metastatic diseases, it is important to further investigate it to find optimal patient-specific surgical methods and their safety. Hence, the topic of this manuscript seems to be on point and can be published after minor revisions after addressing the following concerns.

1.     Table 1 is a list of the previous studies, which should not occupy a big portion of this article. Therefore, it is recommended that Table 1 can be moved to the supplementary materials. 

2.     In Figure 4, following the graphs between 0 and 20 months is hard. Please revise it accordingly.

3.     (Page 15, lines 332-335) This sentence seems to be more suitable for the materials and methods section.

4.     In the discussion section, there are some repetitive sentences that appear in the materials and methods or introduction section. Please remove them to make the discussion brief and straightforward. 

5.     The limitation section is too long. Please make it brief. Also, it is recommended to divide this paragraph into two to include the future plan. 

Author Response

Response to Reviewer 1 Comments

Point 1: Table 1 is a list of the previous studies, which should not occupy a big portion of this article. Therefore, it is recommended that Table 1 can be moved to the supplementary materials.

Response 1: Table 1 has been moved to the supplementary materials and is now labeled as Supplement 1. The previous Supplement 1 is now labeled as Supplement 2, and the previous Table 2 is now labeled as Table 1, Table 3 as Table 2, and Table 4 as Table 3.

Point 2: In Figure 4, following the graphs between 0 and 20 months is hard. Please revise it accordingly.

Response 2: Thank you for this comment. We have revised the coloration of the lines, changed the x-axis intervals to 12 months instead of 20 months, and inserted an x-axis break between 36 and 84 months in order to make the graph more legible.

Point 3: (Page 15, lines 332-335) This sentence seems to be more suitable for the materials and methods section.

Response 3: This sentence was already included in the Materials and Methods section (Supplement 2, previously Supplement 1). We included it in the Discussion section as it pertains to the relatively low emboli rate in our series.

Point 4: In the discussion section, there are some repetitive sentences that appear in the materials and methods or introduction section. Please remove them to make the discussion brief and straightforward.

Response 4: Thank you for your suggestion to make the writing more concise. We have removed some repetitive sentences (lines 284-288, 289-291, 303-305, 357-360, 374-375) to make the Discussion brief and straightforward.

Point 5: The limitation section is too long. Please make it brief. Also, it is recommended to divide this paragraph into two to include the future plan.

Response 5: We have made the limitation section more brief by removing several phrases and sentences (lines 352-353, 356-360, 374-376). We have also divided the paragraph into two and included the future plan (lines 390-393).

Reviewer 2 Report

This is a well written large series about the IMN for fractures of the humerus, based on a single-surgeon database. It could be interesting to know if results are similar in the cohort of patients with multiple myeloma in which a more in deep analysis could be performed. 

Author Response

Response to Reviewer 2 Comments

Point 1: This is a well written large series about the IMN for fractures of the humerus, based on a single-surgeon database. It could be interesting to know if results are similar in the cohort of patients with multiple myeloma in which a more in deep analysis could be performed.

Response 1: Thank you for your comment and interesting suggestion for additional analysis. We compared length of survival and complications (medical, surgical, and overall) between patients with multiple myeloma and patients with other metastatic disease and found no statistical difference. Therefore, we reported this finding by adding the sentence “Nevertheless, there was no significant difference (p>0.05) in perioperative complications or survivorship between those with multiple myeloma and those with metastatic disease” in lines 382-384 of the manuscript. We have also included the specific numerical outputs below for your reference.

Survival

Multiple Myeloma: 11.38 months

Metastasis: 7.35 months

p=0.2025

Medical Complications

Multiple Myeloma: 8

Metastasis: 5

p=0.7667

Surgical Complications

Multiple Myeloma: 3

Metastasis: 4

p=0.7001

Overall Complications

Multiple Myeloma: 11

Metastasis: 10

p=1.00